# Factors Associated with Bone Union Failure After Frozen Autograft Reconstruction in Lower Limb Osteosarcoma

**DOI:** 10.3390/cancers17223601

**Published:** 2025-11-07

**Authors:** Sei Morinaga, Katsuhiro Hayashi, Shinji Miwa, Takashi Higuchi, Hirotaka Yonezawa, Yohei Asano, Hiroyuki Tsuchiya, Satoru Demura

**Affiliations:** 1Department of Orthopaedic Surgery, Graduate School of Medical Sciences, Kanazawa University, Kanazawa 920-8641, Japan; 2Department of Orthopaedic Surgery, Yokohama Sakae Kyosai Hospital, Yokohama 247-8581, Japan

**Keywords:** osteosarcoma, frozen autograft, biological reconstruction, multiple plate, intramedullary nail

## Abstract

**Simple Summary:**

Liquid nitrogen-treated frozen autograft is a distinctive biological reconstruction method developed at Kanazawa University for patients with malignant bone tumors. Although this technique preserves the patient’s own bone and avoids prosthetic replacement, bone union between the frozen graft and host bone remains challenging. We studied 35 patients with osteosarcoma of the lower limb long bones to identify factors influencing bone union. Fixation with intramedullary nails was strongly associated with nonunion, whereas plate fixation, particularly with multiple plates, achieved more reliable healing, supporting plate fixation as the preferred method to reduce the risk of nonunion in frozen autograft reconstruction.

**Abstract:**

**Background/Objectives**: Liquid nitrogen-treated frozen autograft is a biological reconstruction method developed at Kanazawa University for malignant bone tumors. However, nonunion between the treated autograft and host bone remains a complication. In this study, we aimed to identify factors influencing bone union in patients undergoing this procedure for osteosarcoma of long bones in the lower extremities. **Methods**: We retrospectively analyzed 35 osteosarcoma patients (mean age: 18.0 years) with lower limb long bone tumors treated with frozen autograft between 1999 and 2023. Factors assessed included sex, age, tumor location, fixation method (plate or intramedullary nail), technique (pedicle or free freezing), chemotherapy, and bone union. **Results**: Nonunion occurred in 6 cases: 2/25 with plate fixation (8.0%) and 4/10 with intramedullary nails (40%). The mean union time was shorter with plates (5.8 months) than with nails (7.2 months). Intramedullary nail use was significantly associated with nonunion (*p* < 0.05). Among plate fixations, nonunion occurred in 5.6% of multiple plates versus 14.3% of a single plate. **Conclusions**: Intramedullary nail fixation is associated with nonunion in biological reconstructions of long bones, consistent with previous reports. Multiple-plate fixation after frozen autograft with liquid nitrogen for osteosarcoma of the lower limb long bone should be considered.

## 1. Introduction

Osteosarcoma is the most common primary malignant bone tumor of mesenchymal origin, typically arising during the adolescent growth spurt, with a smaller incidence peak also occurring in older adults [1,2,3]. The standard of care consists of wide surgical resection combined with multi-agent systemic chemotherapy. The principal chemotherapy regimens, established in the 1970s, include high-dose methotrexate, doxorubicin, cisplatin, and ifosfamide, with or without etoposide [4,5]. Despite these intensive multimodal strategies, the 5-year overall survival rate has plateaued at approximately 60–70%, and disease-specific mortality remains around 30–40% in most series, with minimal improvement over the past 3 decades [6,7,8]. Therefore, novel approaches in surgical techniques and adjuvant therapies remain essential to improve patient outcomes.

Limb-salvage surgery for malignant bone tumors requires reconstruction of large bone defects using prosthetic, biological, or hybrid techniques [9,10,11,12]. Prosthetic reconstruction remains widely employed; however, its long-term durability is limited, and repeated reoperations are often necessary because of mechanical failure, loosening, or chronic infection [13,14,15]. Biological reconstruction using allografts or autografts sterilized by heat, irradiation, or deep freezing has been established as an alternative strategy [16,17,18,19,20,21]. In particular, tumor-bearing autografts treated with irradiation, pasteurization, or liquid nitrogen are now extensively utilized in many countries for the reconstruction of bone defects following tumor resection.

Frozen autograft using liquid nitrogen is a biological reconstruction technique originally developed at the Department of Orthopedic Surgery, Kanazawa University [18]. This approach preserves bone stock and osteoinductive properties and is currently applied in Japan and other countries, particularly for the treatment of osteosarcoma [22,23,24]. Initially applied for osteosarcoma of the lower limb after wide resection, its indications have gradually expanded to include other malignant bone tumors. Over the past two decades, more than 170 cases have been performed at our institution, demonstrating 5- and 10-year graft survival rates of 83% and 70%, respectively [22]. These results have established the frozen autograft as a reliable limb-salvage option for malignant bone tumors (Figure 1). However, nonunion between the treated graft and host bone remains a significant complication (7.2–26.7%), potentially impairing limb function and requiring additional surgical intervention [25,26].

Therefore, the present study aimed to identify clinical and surgical factors associated with bone union in patients with osteosarcoma of the lower extremities who underwent reconstruction with frozen autografts treated with liquid nitrogen.

## 2. Materials and Methods

A total of 172 patients underwent surgery with frozen autografts at our institution between January 1999 and December 2023. Of these, 43 patients with lower limb osteosarcoma were identified. Patients who experienced local recurrence, infection, or fracture at the surgical site before bone union were excluded. Ultimately, 35 patients with intercalary osteosarcoma of the lower limb long bones who were followed for at least one year were retrospectively analyzed (Figure 2). The mean age at surgery was 18 years (range, 6–62 years).

The factors assessed included sex, age, tumor location, fixation method (plate or intramedullary nail), freezing technique (pedicle or free), chemotherapy, and bone union. Sex was included as an independent variable in the analysis; however, it was not a selection criterion and did not influence the choice of fixation method or chemotherapy schedule. The base protocol of chemotherapy was given as five pre-operative courses of intra-arterial or intravenous cisplatin (120 mg/m^2^) and doxorubicin (30 mg/m^2^/day × 2 days), according to our institutional regimen. An additional six courses of post-operative chemotherapy were administered beginning two to three weeks after surgery, depending on patient recovery and hematologic status. For plate fixation, the effect of one versus multiple plates was also evaluated. Bone union was assessed using plain radiographs, tomography, or CT scans. Union was defined as continuous callus formation at the graft–host junction. Cases that did not achieve union within one year or required revision surgery for nonunion were classified as nonunion.

### 2.1. Surgical Procedure

#### 2.1.1. Free Freezing Technique [18]

For the free freezing procedure, the tumor-bearing bone was excised en bloc with an adequate surgical margin using a microsurgical saw or T-saw. After removal of soft tissue, the intramedullary tumor was curetted to minimize fracture risk during freezing. The specimen was immersed in liquid nitrogen for 20 min, thawed at room temperature for 15 min, and subsequently thawed in 37 °C distilled water with 0.3% iodine solution for another 10 min. The frozen autograft was then reimplanted and fixed to the residual host bone using an intramedullary nail or single/double locking plates.

#### 2.1.2. Pedicle Freezing Technique [27]

The pedicle freezing procedure was applied to intercalary cases in which the adjacent joint was preserved. The tumor-bearing segment was exposed by osteotomy with an adequate surgical margin while maintaining continuity with the joint. Surrounding normal tissues were carefully protected using surgical sheets, cotton padding, and bandages to avoid contamination or thermal injury. The tumor within the bone was curetted to reduce fracture risk during freezing caused by water expansion. The affected bone segment, still attached to the joint, was rotated and immersed in liquid nitrogen for 20 min, followed by thawing at room temperature for 15 min and in distilled water or 0.3% iodine solution for 10–15 min. Reconstruction was then performed in the same manner as the free freezing procedure, using intramedullary nails or plates.

All statistical analyses were conducted using EZR (version 1.54; Saitama Medical Center, Jichi Medical University, Saitama, Japan) [28]. The associations between bone union and other clinical or surgical factors were assessed using Fisher’s exact test. A *p*-value < 0.05 was considered statistically significant.

## 3. Results

### 3.1. Patients’ Characteristics

Among the cohort, 17 patients were male and 18 were female. The histological subtypes included conventional osteosarcoma in 33 cases (osteoblastic, 27; chondroblastic, 3; fibroblastic, 3), parosteal osteosarcoma in 1 case, and small cell osteosarcoma in 1 case. Tumor localization was the femur in 19 cases and the tibia in 16 cases. Fixation methods included plate fixation in 25 patients (single plate: 7, multiple plates: 18) and intramedullary nailing in 10 patients. Reconstruction methods comprised pedicle freezing in 13 cases and free freezing in 22 cases. Chemotherapy was administered in 31 patients. Local recurrence was observed in 7 patients (20%).

The detailed patient characteristics are summarized in Table 1.

### 3.2. Nonunion Analysis

Nonunion occurred in 6 cases: 2 of 25 with plate fixation (8.0%) and 4 of 10 with intramedullary nailing (40%). The mean time to union was shorter in the plate fixation group (5.8 months) compared with the intramedullary nail group (7.2 months). The use of intramedullary nails was significantly associated with a higher risk of nonunion (*p* < 0.05, Table 2). Among plate fixations, nonunion occurred in 1 case (5.6%) with multiple plates, compared to 1 case (14.3%) with a single plate (*p* = 0.49, Table 3).

### 3.3. Cases

Case 1 was a 62-year-old man. Pedicle freezing reconstruction was performed with an intramedullary nail for a conventional osteoblastic osteosarcoma of the left diaphyseal femur. Bone union was not achieved until 1 year and 3 months after surgery, and an iliac bone graft with additional plate fixation was performed (Figure 3).

Case 2 was a 74-year-old woman. Pedicle freezing reconstruction was performed with double plates for a conventional osteoblastic osteosarcoma of the right diaphyseal tibia. Bone union was confirmed 6 months after surgery (Figure 4).

## 4. Discussion

In this study, we investigated factors associated with bone union after reconstruction using liquid nitrogen-treated frozen autografts for osteosarcoma of the lower limb long bones. We found that intramedullary nailing was significantly associated with a higher risk of nonunion, whereas fixation with multiple plates provided more reliable bone healing. These findings highlight the importance of achieving sufficient rotational and mechanical stability when performing biological reconstruction with frozen autografts.

The biomechanical environment of the lower extremity differs substantially from that of the upper limb, which may explain why our findings particularly emphasize the importance of rotational stability. The femur and tibia are exposed to repetitive axial loading and torsional stress during ambulation, generating greater mechanical demands at the graft–host junction. In contrast, upper-extremity reconstructions typically experience lower cyclic loading. Therefore, the advantage of multiple-plate fixation observed in our study may be especially relevant to weight-bearing bones of the lower limb, where stronger stabilization is essential to promote bone union.

Previous studies have demonstrated that frozen autografts treated with liquid nitrogen initially enter a devitalized phase but subsequently undergo gradual revitalization. Xu et al. reported in a rabbit model that no viable cells were detected in the frozen area during the first 1–8 weeks; however, from 12 weeks onward, fibrous tissue, osteoblasts, and vascular ingrowth progressively invaded the necrotic bone, resulting in replacement by newly formed bone through creeping substitution and both endochondral and intramembranous ossification [29]. Similarly, Araki et al. evaluated the viability of frozen autografts in patients using 99 mTc-MDP scintigraphy and observed that tracer uptake temporarily decreased after surgery but subsequently increased, with normalization achieved in 95–97% of cases within 60 months. These findings suggest that frozen autografts undergo an initial devitalized state followed by progressive revitalization over time [30].

Intramedullary nailing has been shown to cause a transient reduction in intramedullary blood flow. Reaming and nailing embolize marrow contents into the systemic circulation and decrease blood flow to the bone and cortex by 30% to 80% [31]. Similarly, in a rat model, blood flow in the diaphyseal femur immediately after reaming was reduced to about one third of that in the intact femur [32]. In addition, intramedullary nails may not always provide adequate stability. Mechanical causes of fixation failure and nonunion include undersized nails, rotational instability, malalignment, or comminution, as reported in several studies [33,34,35]. Rotational instability, in particular, has been identified as a major cause of nonunion after intramedullary nailing. Taken together, these factors suggest that intramedullary nailing, by both reducing intramedullary blood flow and compromising mechanical stability, may impair the revitalization process of frozen autografts. In the present study, intramedullary nail use was identified as a risk factor for nonunion in liquid nitrogen reconstruction. Previous reports have similarly shown that intramedullary nail fixation is a primary risk factor for nonunion [36,37], largely because intramedullary nails are believed to lack sufficient rotational stability in intercalary reconstruction. In the present study, nonunion occurred in 5.6% of cases treated with multiple plates, compared with 14.3% of those treated with a single plate. Although this difference was not statistically significant, it is likely attributable to the limited sample size inherent in studies of osteosarcoma involving long bones of the lower extremity. Intercalary allograft augmented with plate fixation has been described as a reliable solution after diaphyseal tumor resection, in which dynamic compression plates were used to fix the allograft to the host bone, usually two in the lower extremity [38]. Similarly, in the Capanna technique, plates were applied on each side of the bone, 180 degrees opposite each other, consisting of multiple plates spanning each host–allograft junction [39]. Furthermore, in frozen autograft reconstruction, the frozen autograft is stabilized with double or triple locking plates [40]. Taken together, these findings indicate that intercalary reconstruction using frozen autografts treated with liquid nitrogen is now commonly performed with multiple plate fixation to ensure adequate stability and reliable bone union. In addition, a multi-institutional study from the Japanese Musculoskeletal Oncology Group reported that the absence of events and graft removal correlated with improved limb function, whereas composite grafts correlated with reduced limb function [37]. These results emphasize that postoperative events such as nonunion or graft removal are closely linked to functional decline, underscoring the critical importance of achieving bone union after intercalary reconstruction for malignant bone tumors.

Osteolytic bone tumors are generally not recommended for liquid nitrogen-treated frozen autograft reconstruction; however, if osteolysis is not advanced, polymethyl methacrylate (PMMA) or bioactive bone cement may be used to augment reconstruction [25]. In this setting, liquid nitrogen-treated autografts are considered most appropriate for mild osteolytic or osteoblastic tumors, but not for severely osteolytic lesions, particularly osteosarcoma.

Cryotreatment, such as frozen autograft, can induce the abscopal effect, a secondary immunologic phenomenon. This effect describes the regression of tumors at sites distant from the primary lesion following local radiotherapy. First reported by Mole and colleagues in 1953, it refers specifically to unexpected shrinkage of untreated metastases after radiation exposure [41,42]. Radiotherapy can trigger the abscopal effect through immunogenic cell death, releasing tumor antigens and danger signals such as HMGB1 and calreticulin, which activate dendritic cells and prime cytotoxic CD8^+^ T cells via the cGAS–STING/type I interferon pathway. These immune responses enable systemic recognition and clearance of metastases, although the effect may be counterbalanced by radiation-induced immunosuppressive mechanisms, including regulatory T cells (Tregs) and myeloid-derived suppressor cells (MDSCs) [43]. The abscopal effect of cryotreatment has been documented in patients with metastatic cancers [44,45]. Recent evidence suggests that combining cryotreatment with immunotherapy could potentiate the abscopal response [46,47,48,49,50,51,52]. In addition, growing evidence indicates that cryoablation offers unique immunological benefits, such as rejection of secondary tumor challenges [53,54,55,56].

Beyond local tumor control, liquid nitrogen-based reconstruction appears to stimulate systemic antitumor immunity consistent with an abscopal-like effect. Clinically, regression of multiple lung metastases has been reported after cryotreatment for renal cell carcinoma bone metastasis, accompanied by marked increases in interferon-gamma (IFN-γ) and interleukin-12 (IL-12) [57]. A series of total en bloc spondylectomy with frozen autografts further showed significant postoperative rises in IFN-γ (mean 275% at 3 months) and IL-12 (486% at 3 months), supporting a systemic immune response [58]. These findings suggest that the immunologic benefits of cryotreatment may extend beyond local control, offering a novel therapeutic strategy. With further refinement and clinical application, this approach holds promise to improve the long-stagnant prognosis of osteosarcoma.

Liquid nitrogen-treated autografts enable true biological reconstruction: cryogenic devitalization preserves the graft’s collagenous matrix and osteoinductive milieu, allowing vascular ingrowth and osteogenesis, i.e., creeping substitution and subsequent revitalization. Recent cohort data for intercalary reconstructions reinforce this concept, reporting favorable union and durable outcomes consistent with progressive biologic incorporation [25]. Miwa et al. reported 5- and 10-year graft survival rates of 83% and 70% after liquid nitrogen-treated autograft reconstruction, outcomes considered superior to other biological reconstructions [16,22,59,60,61,62,63,64,65,66]. Collectively, these findings justify positioning liquid nitrogen-treated autografts as a cost-effective, durable option when long-term physiologic remodeling is desired. While liquid nitrogen-treated autografts enable lasting biological incorporation, two persistent concerns remain: degeneration of articular cartilage after osteoarticular reconstruction [67] and delayed union or nonunion at host–graft junctions [68,69]. However, the retrospective design and relatively small cohort size of this study may limit its generalizability. Therefore, multi-institutional validation with larger sample sizes is warranted to confirm these findings. Because of the limited number of nonunion events, multivariate analysis was not performed to avoid model overfitting; thus, the univariate results should be interpreted with caution. Although sex was included as a variable, no significant association with bone union was observed. In a previous multicenter study on biological reconstruction, male sex was reported as a significant risk factor for nonunion [37]. Larger multicenter osteosarcoma cohorts may further clarify potential biological or biomechanical sex-related effects on graft healing. Another limitation of this study is that bone union was primarily assessed using plain radiographs. In cases with double-plate fixation, hardware superimposition may obscure the bridging callus, potentially leading to underestimation or delayed recognition of union. Future studies incorporating CT-based evaluation may allow for more precise assessment of graft–host integration. Further investigations are required to address these limitations.

## 5. Conclusions

The risk factor for nonunion after reconstruction with frozen autograft for lower limb long bone osteosarcoma was intramedullary nail use. To achieve successful bone union, fixation with multiple plates is recommended in this procedure.

## Figures and Tables

**Figure 1 cancers-17-03601-f001:**
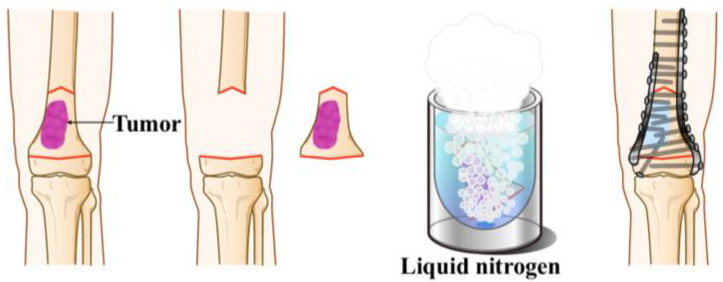
**Schematic diagram of frozen autograft reconstruction with liquid nitrogen (Free Freezing Technique).** The tumor-bearing bone segment is first excised en bloc with an adequate surgical margin. The resected bone is then frozen in liquid nitrogen at −196 °C for 20 min to eradicate all tumor cells, followed by thawing at room temperature and washing in distilled water. The treated bone is reimplanted anatomically and fixed with one or more plates or an intramedullary nail.

**Figure 2 cancers-17-03601-f002:**
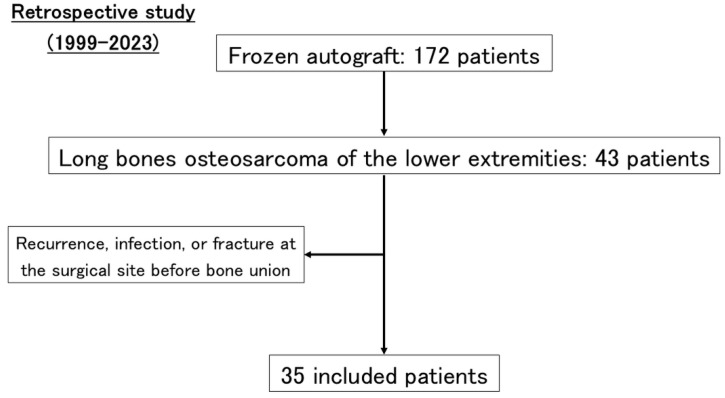
**Flow diagram illustrating the patient selection process.** Among 172 patients who underwent liquid nitrogen-treated frozen autograft reconstruction at Kanazawa University between 1999 and 2023, 43 cases involving long bones of the lower extremities were identified. After excluding patients who developed local recurrence, infection, or fracture at the surgical site before bone union, 35 patients were ultimately included in the analysis.

**Figure 3 cancers-17-03601-f003:**
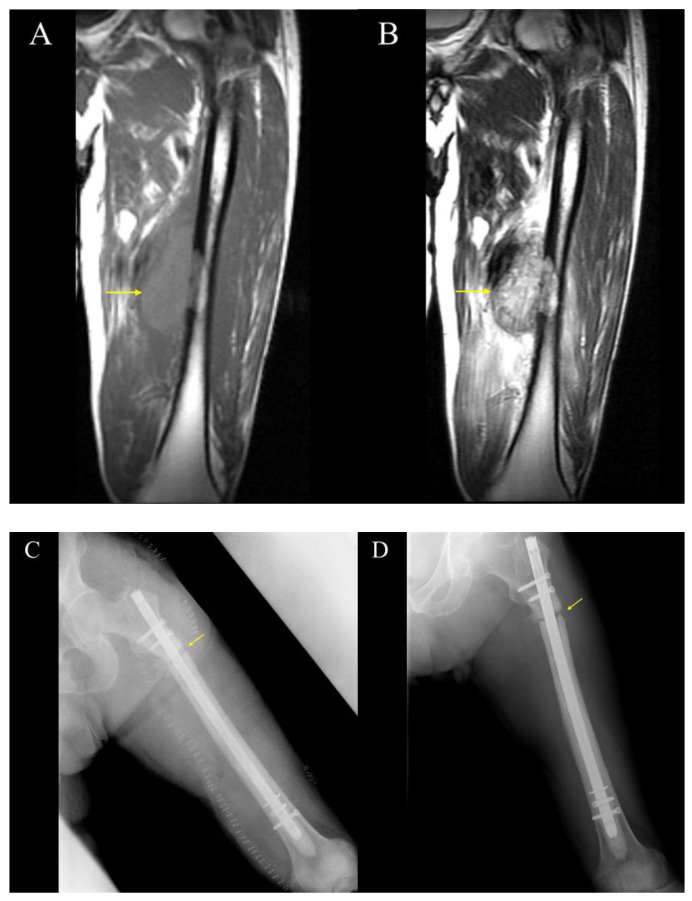
**The images of Case 1.** (**A**) Preoperative T1-weighted MRI; (**B**) Preoperative T2-weighted MRI; (**C**) Immediate postoperative X-ray (Day 0); (**D**) X-ray at 1 year and 3 months after surgery demonstrating nonunion at the osteotomy site. The yellow arrows mean a tumor in (**A**,**B**), and osteotomy line in (**C**,**D**).

**Figure 4 cancers-17-03601-f004:**
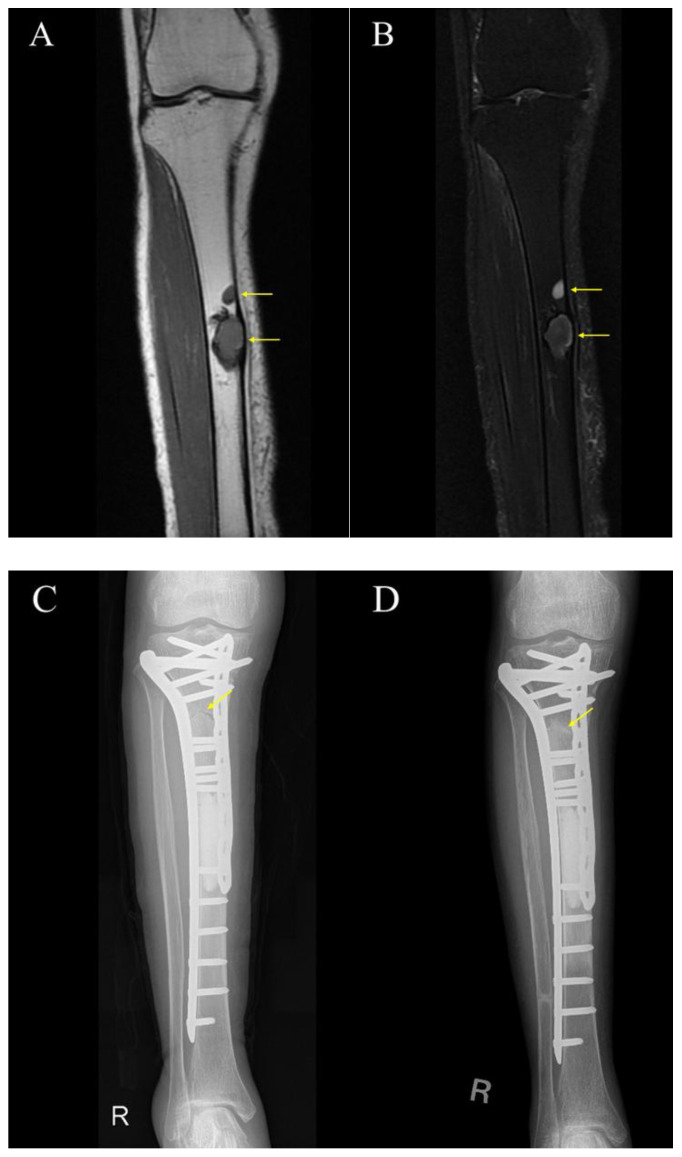
**The images of Case 2.** (**A**) Preoperative T1-weighted MRI; (**B**) Preoperative STIR-weighted MRI; (**C**) Immediate postoperative X-ray (Day 0); (**D**) X-ray at 6 months after surgery demonstrating union at the osteotomy site. The yellow arrows mean tumors in (**A**,**B**), and osteotomy line in (**C**,**D**).

**Table 1 cancers-17-03601-t001:** **Patients’ characteristics.**

Characteristic		Number
Mean age		18 (6–62)
Sex	Male	17
	Female	18
Histology	Conventional osteosarcoma	33
	(Osteoblastic)	27
	(Chondroblastic)	3
	(Fibroblastic)	3
	Paroosteal osteosarcoma	1
	Small cell osteosarcoma	1
Location	Femur	19
	Tibia	16
Fixation method	Plate	25
	Nail	10
Freezing method	Pedicle freezing	13
	Free freezing	22
Chemotherapy	+	31
	-	4
Bone union	+	28
	-	7

**Table 2 cancers-17-03601-t002:** **Statistical analysis for bone union.** The use of intramedullary nail with frozen autograft was a significant risk factor for nonunion.

Variables		Bone union		Fisher’s exact test
		+	-	*p* value
gender	male	14	3	1
	female	14	4	
age	<20	24	4	0.13
	≥20	4	3	
Localization	Femur	17	2	0.21
	Tibia	11	5	
Reconstruction method	Plate	23	2	0.043
	Nail	6	4	
Freezing method	Pedicle	10	3	1
	Free	18	4	
Chemotherapy	-	3	1	1
	+	25	6	

**Table 3 cancers-17-03601-t003:** **Statistical analysis for bone union regarding the number of plate fixations.** Single plate fixation tended to result in nonunion compared to multiple fixation.

Variable		Bone union		Fisher’s exact test
		+	-	*p* value
Plate	Single	6	1	0.49
	Multiple	17	1	

## Data Availability

The data presented in this study are available on reasonable request from the corresponding authors.

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
