# Peer review of "Factors Associated with Bone Union Failure After Frozen Autograft Reconstruction in Lower Limb Osteosarcoma"

_cancers, 2025, doi:10.3390/cancers17223601_

Round 1
Reviewer 1 Report
Comments and Suggestions for Authors
This was a nice retrospective review that was able to achieve statistically significant results with a modest sample size. There are a few major clinically-relevant take-aways regarding how to fixate frozen autografts that are useful to readers: plate is greater than nail, 2 plates are better than 1, and rotational stability is critical to achieve. My only recommendation would be to elaborate on how these results may be unique to lower extremity reconstruction, since forces across lower and upper extremities may differ. Additionally, there is no introductory sentences in the Discussion-- it may be nice to include a summary sentence that reminds the reader of the goals of the study, and key findings, before jumping into the heavy details of the Discussion. Finally, a weakness not mentioned was the limitations of using xrays to define union-- with two plates, the ability to view bridging bone at the osteosynthesis site may be limited.
Author Response
We appreciate these insightful comments. We have added a new paragraph in the Discussion section (page 7, lines 183–190) explaining that lower-limb bones are subject to greater axial and torsional loading during gait than upper-limb bones. We note that this biomechanical environment may explain the stronger association between multiple-plate fixation and successful union in the lower extremities. We have also revised the opening of the Discussion to include 2 sentences summarizing the purpose of the study and the principal findings (page 7, lines 177–182).
We have added a statement in the Discussion (page 9-10, lines 289–294) acknowledging that evaluation by plain radiography may underestimate union in double-plate fixation, and that CT assessment could provide more accurate evaluation in future studies.
Reviewer 2 Report
Comments and Suggestions for Authors
I enjoyed reading this well written and thoughtful manuscript. I have very little to add. The manuscript is well referenced and the figures and tables are of excellent quality.
- Could more detail be provided regarding the chemotherapy schedules used.
- Detail regarding the recurrence rates could be incorporated.
Author Response
We appreciate your suggestion. We have now added a detailed description of our institutional chemotherapy protocol in the Materials and Methods section (page 3, lines 102–106). Briefly, five pre-operative courses of intra-arterial or intravenous cisplatin (120 mg/m²) and doxorubicin (30 mg/m²/day × 2 days) were administered, followed by six post-operative courses beginning two to three weeks after surgery.
We have also added the local recurrence rate (20%) to the Results section (page 4, lines 143–144), indicating that 7 patients experienced local recurrence.
Reviewer 3 Report
Comments and Suggestions for Authors
Suggestions and Comments for the Authors
The manuscript presents a valuable contribution by reporting clinical cases using a novel technique developed at Kanazawa University. The data are promising, and the overall structure is sound. However, several key areas require clarification and expansion, as follows:
- In the first paragraph of the introduction, please include epidemiological data on osteosarcoma, specifically the mortality rate, to provide context for the significance of your study.
- Since the technique was developed at your institution, it would strengthen the manuscript to include more detailed statistical discussion about its development and application. Additionally, consider adding a schematic diagram of the method and its outcomes in the introduction to enhance reader understanding.
- The role of sex in the methodology is currently unclear. Please clarify whether sex was considered in patient selection, treatment response, or outcome analysis. If this factor has been addressed in previous studies, cite them in the introduction. Otherwise, discuss its relevance and implications in the discussion section.
Best of luck with your revisions and future research.
Author Response
1. In the first paragraph of the introduction, please include epidemiological data on osteosarcoma, specifically the mortality rate, to provide context for the significance of your study.
Response: We appreciate this helpful suggestion. We have added global epidemiological data on osteosarcoma survival and mortality to the first paragraph of the Introduction (page 2, lines 48–49) to better emphasize the clinical relevance of this study.
2. Since the technique was developed at your institution, it would strengthen the manuscript to include more detailed statistical discussion about its development and application.
Additionally, consider adding a schematic diagram of the method and its outcomes in the introduction to enhance reader understanding.
Response: Thank you for this insightful recommendation. We have expanded the Introduction (page 2, lines 65–70) to describe the history of liquid-nitrogen frozen autografts at Kanazawa University and included the number of institutional cases and long-term graft-survival outcomes. Moreover, we have added a new schematic diagram (Figure 1) illustrating the procedure and expected results to enhance reader comprehension.
3. The role of sex in the methodology is currently unclear. Please clarify whether sex was considered in patient selection, treatment response, or outcome analysis. If this factor has been addressed in previous studies, cite them in the introduction. Otherwise, discuss its relevance and implications in the discussion section.
Response: We appreciate this insightful comment. We have clarified in the Materials and Methods section that sex was included as an independent variable in the analysis but was not used as a criterion for patient selection or treatment allocation (page 3, lines 100–102). In the Discussion section, we have added a statement noting that no significant sex-related differences in bone union were observed in our cohort. However, the previous study has reported that male sex can be a risk factor for nonunion after biological reconstruction. We have now discussed this point and suggested that larger multicenter analyses may help further elucidate potential biological or biomechanical sex-related influences on graft healing (page 9, lines 285–289).
Round 2
Reviewer 3 Report
Comments and Suggestions for Authors
Well response. Good luck!